# Precision Cutting of the Molds of an Optical Functional Texture Film with a Triangular Pyramid Texture

**DOI:** 10.3390/mi11030248

**Published:** 2020-02-27

**Authors:** Huang Li, Zhilong Xu, Jun Pi, Fei Zhou

**Affiliations:** College of Mechanical and Energy Engineering, Jimei University, Xiamen 361021, China; li_huang6002@163.com (H.L.); Pi_jun@163.com (J.P.); zf18850043071@163.com (F.Z.)

**Keywords:** optical functional texture film, area ratio of retro-reflection, triangular pyramid texture, mold of light trapping film

## Abstract

Based on an analysis of the precision and preparation technology of an optical texture film with a triangular pyramid texture, the technical requirements of the original mold were determined, and precision shaping planning technology was adopted to process the original mold. The shape error of the optical texture mold of the triangular pyramid was assessed by defining the area ratio of the retro-reflection. The influence of the tool nose radius and exit burr on the area ratio of the retro-reflection were analyzed. By optimizing the cutting tools, cutting materials and cutting boundaries, a five-axis ultra-precision machining system was used to plan the triangular pyramid structure with a base length of 115 µm and an included angle between two sides of 70.5°. The experimental results indicate that the dimension error of the triangular pyramid element is less than 1 µm, the angle error of the included angle between two sides is less than 0.05°, and the average roughness of the side of the triangular pyramid can reach 9.2 nm, which satisfies the processing quality requirements of the triangular pyramid texture mold.

## 1. Introduction

An optical functional texture film is a surface microarray film that can change the path of light propagation [1]. With its exceptional light conductivity and self-cleaning function, this texture film has attracted the attention of many researchers. With improvements in ultra-precision machining technology, the preparation of the optical functional texture of surfaces has become a popular research topic in precision machining. There are two classes of methods in preparing an optical functional texture film. One type of method involves directly preparing the optical functional texture on the surface of the work piece with technology, such as plasma etching [2] and lithography [3]. The other is to obtain the optical functional texture via embossing technology, such as hot embossing [4,5] and ultraviolet (UV) imprinting [6,7]. In 2015, Rahman [8] et al. published in Nature Communications that three types of nanoscale tapered light-trapping textures were directly prepared on the surface of a poly(methyl methacrylate) (PMMA) thin film via plasma etching. The tapered light-trapping texture ranged from 10–70 nm in diameter, and the light absorption rate of the smooth crystal silicon cells was improved. Diao et al. [9] prepared moth-eye nanostructures with high transmission and anti-reflection on optical substrates via block copolymer micelle lithography (BCML) and reactive ion etching. Ion etching or photolithography can be used to directly prepare the optical functional texture with a high surface quality and machining accuracy. However, these methods demonstrate complex processing with a low processing efficiency and high manufacturing costs; thus, they are unsuitable for large-scale production and have remained only in the experimental research stage.

An optical functional textured film was prepared by using embossing technology. First, a mold with texture was processed, and then the mold texture was embossed on the surface of an optical film [10]. This method has a high manufacturing efficiency and a low production cost and can be employed in mass production. According to the microstructural dimensions of the mold texture, the texture can be divided into nanoimprints and microimprints. The surface microstructure of the nanoimprint is on the nanoscale. It is difficult to ensure the integrity of the surface microstructure during the replication process of a nanoimprint; thus, the optical properties of the optical functional texture will be affected. Optical functional microstructures on the micrometer scale have high precision and are commonly used in all types of light-trapping films and reflective films. Zhang et al. [11] processed a high-precision Fresnel lens texture on the cylindrical surface of a roller die and replicated a Fresnel lens array on the surface of polyethylene terephthalate (PET) film via a UV embossing method to obtain a high-precision concentrated functional surface texture. Jiang et al. [12] replicated holographic images on a high-precision roll die on the surface of a biaxially-oriented polypropylene (BOPP) film in a large area by means of hot embossing, and obtained a diffraction grating texture with superior accuracy.

To obtain improved optical properties, the precision of the surface texture film is required to be high, and the core technology using the embossing process is used to prepare a high-precision embossing mold [13]. Ultra-precision cutting with a diamond tool can produce microstructural surfaces with roughness values less than 10 nm [14]. Brinksmeier et al. [15] proposed a diamond microchiseling (DMC) technique for machining hexagon cube corner arrays with a structure size of 100 µm on the surface of a nickel-plated mold, and its side surface roughness reached *S*a = 3 nm, meeting the quality requirements of a mirror reflection. Kim et al. [16] used a single-crystal diamond tool to produce a micropyramid array with a height of 25 µm from the surface of the copper-plated mold, and its lateral roughness reached 14.8 nm.

Scholars have carried out research on optical functional structures at sub-wavelength and sub-millimeter scales. By using the wave theory of light analysis, it can be concluded that composite optical functional structures in the sub-wavelength range can reduce the refractive index of thin film materials, thereby reducing the reflection loss of solar cells [17]. Rahman et al. [8] laminated a layer of a flat film to the surface of crystalline silicon and then processed a fine conical texture 5 nm in height on the flat film using the ultra-precision ion etching method, which decreases the reflection loss of the composite structure crystalline silicon cells to below 1%. However, the area of the prototype is only 0.5 cm^2^ due to the prohibitively high cost. It is currently still difficult to produce large-area optical functional textured films using the ion etching technique. In addition, the nanoembossing technique allows for the fabrication of nanoscale optical functional textured films over a large area, but the fabricated composite laminate structure of crystalline silicon cells only achieves mediocre light performance because of the low fabrication accuracy of the structure. Han et al. [18] laminated PVC-textured films to the smooth surface of crystalline silicon cells using the hot embossing method, which reduced the reflection loss of the crystalline silicon cells from 30% to approximately 15%. Amalathas et al. [19] fabricated a regular inverted pyramidal textured film onto the smooth surface of the crystalline silicon cells at the nanoscale using the ultraviolet (UV) imprinting embossing technique, which lowered the reflection loss of the smooth surface crystalline silicon cells to approximately 14%.

Based on geometrical optics analysis, sub-millimeter-scale light functional textured films can effectively reduce the reflection loss of solar cells. Dottermusch [20] et al. prepared sub-millimeter-scale tapered textures on monocrystalline silicon solar cells, which can reduce the reflection loss to about 8%. Our previous research [21] also found that by laminating a sub-millimeter-scale optical functional texture film to a smooth surface crystalline silicon cell, as shown in Figure 1, the reflection loss of the cell can be reduced to 13%. The embossing process can be used to prepare sub-millimeter-scale optical functional textured films with large area and low cost, and the dimensional accuracy and surface quality of sub-millimeter-scale microstructures can be better controlled. The prepared films have good optical properties.

The processing accuracy of the original master mold has a great influence on the optical performance of the sub-millimeter-scale light functional textured film. In order to further reduce the reflectivity of the light functional textured film, it is an effective method to improve the processing accuracy of the mold. This paper studies how to improve the processing accuracy of the mold, and analyzes the effect of processing accuracy on the performance of the light functional textured film.

The innovations of this paper can be summarized as follows: (1) The influence of ultra-precision machining parameters of micrometer-scale microstructures on the accuracy of the original mold is studied; (2) a new parameter is defined to evaluate the effect of processing errors on the performance of optical functional textured films.

In this paper, according to the technical requirements of the light trapping film with a triangular pyramid, the processing precision of the original mold of the light trapping film (LTF) was formulated. The effects of cutting tools and cutting parameters on the quality of the original mold were experimentally studied. The area ratio of the retro-reflection was used to analyze the influence of the tool nose radius and exit burr on the effective optical functional area of the textured film, laying a foundation for the subsequent preparation of high performance light functional textured film.

## 2. Technical Requirements for the Light-Trapping Texture of the Triangular Pyramid

### 2.1. Optical Functional Textured Film for Light-Trapping

A light-trapping film can improve the absorption rate of its composite devices by modifying its surface microstructure. When the light hits the upper surface of the triangular pyramid texture, it can be refracted into the film several times, thus improving its transmittance. When the light is incident from the subsurface of the triangular pyramid texture film, it will be reflected from the inner surface of the triangular pyramid several times, so the light can be retro-reflected in parallel. To study how to improve the absorption rate of crystalline silicon cells, a research group [21] proposed a microscale triangular pyramidal trapping film and optimized its structural parameters, as shown in Figure 2. The light-trapping texture film is the basic unit of the two symmetrical pyramid structures. The individual right in triangular pyramid structure is the underside of the three sides of an isosceles triangle triangular pyramid. A single pyramid structure is a triangular pyramid whose base is an equilateral triangle and the three sides are isosceles triangles, and its surface triangular pyramid microstructure parameters are as follows: The bottom side *a* = 115 µm, and the included angle between two sides *β* = 70.5. To prevent diffuse reflection of light on the optical functional texture surface, the side surface roughness of microtriangular pyramid must be less than 20 nm.

### 2.2. Light-Trapping Film Mold

#### 2.2.1. Precision of the Size and Surface

Embossing technology can be used to emboss the optical functional texture on the surface of the high-transmittance film. The embossing process is as follows: (1) The original mold of the positive pyramid texture is prepared via ultra-precision machining technology; (2) then, the working mold for replicating the inverted pyramid texture is prepared via precision electroforming processes [22]; and (3) finally, the positive pyramid texture is prepared on the surface of the highlighted transmittance film via hot embossing or UV imprinting. As various errors of the original mold are directly transmitted to the optical functional texture film, high requirements regarding the machining accuracy of the original mold are imposed [23,24,25]. According to the dimensional precision and surface roughness of the light-trapping film, via combination with the precision of electroforming and embossing, the size and surface precision of the microtriangular pyramid of the mold is determined as follows: The bottom side is *a* = 115 ± 3 µm, the included angle between two sides is *α* = 70.5° ± 0.05°, the surface roughness is *Ra* = 10 nm, and the dimensional deviation is less than 3%. To achieve such a high machining quality, the forming process can effectively duplicate the shape of the cutter and ensure the dimensional accuracy of the finished product.

#### 2.2.2. Shape Precision

When cutting the mold with the triangular pyramid texture, because the rounded corners of the tool nose will produce an arc transition at the bottom of the triangular pyramid, and burr will be generated at the edge of the triangular pyramid during processing, these shape errors will be copied to the light trapping film with triangular pyramid, affecting the effective optical functional area of the triangular pyramid texture film. As shown in the Figure 3a, the shadow area of the bottom of the unit texture of the mold and the orthographic projection of edges are invalid optical functional areas, and the width of the triangular pyramid edge burr projection is *W_b_*. Figure 3b shows that the arc radius *r_ε_* of the tool edge is the arc-invalid region of the *EF*. *k_s_* is defined as the ratio of the effective optical functional projection area of the triangular pyramid of the element to the area of the triangular pyramid bottom, i.e., *k_S_* is,
(1)$$k_{s}=1-\frac{\sum_{i=1}^{3n}a_{i}\times r_{\varepsilon }\cos\frac{\alpha }{2}+\frac{\sqrt{3}}{3}\times \frac{\sum_{i=1}^{3n}a_{i}}{3n}\times \sum_{i}^{3n}W_{bi}}{\frac{\sqrt{3}n}{4}{\left(\frac{\sum_{i=1}^{3n}a_{i}}{3n}\right)}^{2}}$$
where *r_ε_* is the tool nose radius, *α* is the included angle between two sides, *W_b_* is the projection width of the triangular edge burr, *a* is the bottom length of the unit triangular pyramid, and *n* is the number of triangular pyramids.

## 3. Cutting Experiment of the Mold

### 3.1. Equipment and Cutting Tools

To avoid the machining errors caused by a change in the cutting depth of the tool, a 5-axis ultra-precision machining system (Moore Nanotech 350F, Moore Nanotechnology Systems Ltd., Swanzey, NH, USA) is adopted as the shaping method for planning. Its repeat positioning accuracy is 0.3 µm, and the positioning error has little effect on the size of the triangular pyramid. As shown in Figure 4a, the work piece is installed on the vacuum clamp of the Z-axis, and a diamond tool with a large tool nose radius and a small tool nose radius is clamped at the left, and right in the tool holder, respectively. The datum plane of the texture is cut by a diamond tool with an enormous arc radius at the tip to eliminate the error of the datum plane caused by the installation. After obtaining the finishing datum plane, the tool holder rotates 180° about axis B, the diamond tool nose processed with a V-groove is aligned with the work piece reference plane, and the principal cutting movement is conducted along the Y-axis. By stepping along the Z-axis of the tool holder, the stepper deep feeding movement of each V-groove is realized. The X-axis stepping motion can produce all parallel V-grooves in the C1 direction on the work piece surface. As shown in Figure 4b, when all V-grooves in the C1 direction are processed, the work piece rotates 60° around the C-axis to process the V-grooves in the C2 direction. When all V-grooves in the C2 direction are processed and the V-grooves in the C3 direction are processed by rotating −120° around the V-axis, a complete triangular pyramid structure can be generated. By setting the diamond tool prior to the experiment, the tool tip of the three cutting directions can be nearly intersected at one point, and the error can meet the processing accuracy of the triangular pyramid texture.

The forming method is adopted for precision planning. The angle of the forming planning tool nose is the same as that of the included angle between the two sides of the triangular pyramid texture. Selecting the rake angle of the tool as 0° can reduce the influence of the tool shape on the shape precision of the forming method. In fine machining, the tool edge radius has a great influence on the machining quality [26]. Studies by Wu et al. [27] showed that the height of the burr increases as the tool edge radius increases. Based on the strength and wear rate of the tool, the tool edge radius of the sharp diamond tool is 0.1 µm. The precision of the forming surface can be improved via ultra-precision grinding of the diamond cutting edge. The increase in the tool nose radius will increase the area of the arc transition zone, thus affecting the area ratio of the retro-reflection of the original mold. Considering the wear rate of the diamond tool nose, the tool nose radius is selected to be 0.1 µm. The geometric parameters of the forming planning tool are reported in Table 1.

To improve the surface accuracy of the triangular pyramidal texture mold, single-crystal diamond is selected as the tool material. The work piece material is composed of brass, which includes 60% copper (Cu), 0.7% tin (Sn) and 39.3% zinc (Zn), and its physical properties are given in Table 2. The work piece size is 20 mm × 20 mm × 40 mm. The friction coefficient *u* = 0.2 between the diamond tool and brass was measured via a friction test. During the process of cutting, spray cooling is adopted at the top of the diamond tool to remove the cutting heat generated during the processing process, and the air-conditioning cooling system of the machining system is turned on to maintain the cutting environment temperature within ±0.1 °C to prevent thermal deformation.

### 3.2. Selection of the Cutting Parameters

The maximum moving speed of the Y-axis of the selected 5-axis ultra-precision machining system is 1500 mm/min, and the cutting speed is 1000 mm/min, considering the stability and flutter of the machine tool and other factors. According to the merchant [28], the expression of the principal cutting force can be derived from Equation (2),
(2)$$F_{y}=\frac{\tau_{s}a_{w}d_{c}\cos(\beta_{e}-\gamma )}{\sin\phi \cos(\phi +\beta_{e}-\gamma )}$$
where *τ_s_* is the shear stress, *a_w_* is the cutting width, *d_c_* is the uncut chip thickness, *β_e_* is the frictional angle, *γ* is the rake angle, and *Φ* is the shear angle.

The Z-axis cutting depth was cut using stepper cutting, as shown in Figure 5. In the cases of other cutting parameters, with an increase in uncut chip thickness *dc*, the principal cutting force and burr height increase gradually [29]. Considering that the diamond tool nose radius and the cutting-edge radius are both 0.1 µm, the principal cutting force is large, which may cause the tool to be microscopically cracked. It is calculated using formula (2) that the uncut chip thickness *dc* should not exceed 6 µm. According to the angle included between the two sides of the unit triangular pyramid, the step depth *ΔZ1* of the tool along the Z-axis during roughing is 10 µm. When finishing, the cutting depth is reduced, the surface quality is better, and the burr size is smaller. However, as the cutting depth is further reduced, the cutting process changes to a plowing process, no chip is generated, and the surface quality worsens [27]. Under a constant cutting depth, the smaller the cutting speed is, the larger the shear strain *ε* is, and it is more likely that a plowing phenomenon occurs [30].

To avoid surface quality degradation caused by the phenomenon of plowing, orthogonal cutting tests under different uncut chip thicknesses were performed, and plowing phenomena were observed. It is concluded that the plowing phenomenon begins when the uncut chip thickness *dc* starts at 1 µm, and the surface roughness begins to deteriorate. The step depth *ΔZ2* of the finishing tool along the Z-axis is 2 µm. According to the length of the bottom edge of the triangular pyramid and the included angle between two sides, the height of the triangular pyramid is 47.2 µm. For each V-groove, the step depth of the tool along the Z-axis is 10 µm, 10 µm, 10 µm, 10 µm, 5.2 µm, and 2 µm.

### 3.3. Quality Inspection of the Mold

#### 3.3.1. Surface Roughness

The surface roughness of the triangular pyramid texture mold will directly affect the optical function of the light-trapping film. A Mitutoyo Formtracer c-5000 (Mitutoyo, Kawasaki, Japan) stylus profilometer was used to measure the surface roughness Ra of the triangular pyramidal mold. As shown in Figure 6a, the work piece is mounted on a rotary table and the work piece is placed obliquely to ensure that the side of the triangular pyramid is in a horizontal position, thereby reducing the measurement error of the roughness. To evaluate the surface quality of the triangular pyramidal texture unit at different processing times, three different positions are selected as the sampling area, as shown in Figure 6b: The diamond tool cut-in area 1, the middle cut 2 and the diamond cutter cut-out area 3. As shown in Figure 6c, each of the sampling regions measures six triangular pyramid units, and each pyramid unit measures the surface roughness values of the middle and lower portions of the three sides. The roughness curve obtained on the typical side is shown in Figure 6d. The surface roughness Ra is up to 8 nm. The roughness measured on 54 sides is averaged, as reported in Table 3, and the average surface roughness Ra is up to 9.2 nm, meeting the surface roughness requirements of the original mold.

#### 3.3.2. Dimensional Accuracy

The triangular pyramid microstructure was measured via laser scanning confocal microscopy (VK-X100K, KEYENCE, Osaka, Japan). The length of the bottom edge of 18 triangular pyramids and the height of 6 triangular pyramids were measured in the sampling area at the three positions in Figure 6b. During the cutting process of the mold, due to the plastic deformation of the material and the existence of processing errors, the processed optical functional texture is not perfect. There are often defects, such as burrs and deformations, and the purpose of this article is to minimize these defects. The average value and deviation were calculated, as reported in Table 3. It can be concluded that the triangular pyramid microstructure size error is approximately 1 µm, the maximum dimension deviation is only 2.82% (less than the required 3%), and the included angle between two sides in the three directions is less than 0.05°, compared with the technical requirement of 70.5°. The actual machining contour of the included angle between two sides is highly consistent with the ideal machining contour. The dimensional accuracy of the triangular pyramid unit in the middle portion of the work piece is higher, which may be the higher stability of the central region.

#### 3.3.3. The Evaluation of the Area Ratio of the Retro-Reflection

As shown in Figure 7a, in the six triangular pyramid elements in the sampling area, the six inner edges of the contrast are found; the shape integrity of the edges 1, 2 and 3 is always better than that of the edges 4, 5 and 6. The possible reasons are analyzed as follows. When cutting in the C1 direction, the cutting is continuous, the cutting state is stable and the six edges are no longer formed. When cutting in the C2 direction, the cutting is no longer continuous; rather, it is interrupted, forming edges 1 and 4, in which edge 4 is formed by the tool at the cut-out and edge 1 is formed by the tool at the cut-in. When cutting in the C3 direction, edges 2, 3 and 5, 6 are formed, in which edges 5 and 6 are formed at the cutting edge of the tool and edges 2 and 3 are formed at the cutting point of the tool. Since the edge of the cut-out can easily form the exit burr, edges 1, 2, and 3 at the cut-in are better than edges 4, 5, and 6 of the cut-out. The *W*_b_-generated burr projection width on edges 4, 5 and 6 was an invalid optical functional area.

Through the above analysis, it is found that there are 6 edges for the texture unit, and the number m of the edges of the exit burrs is only 3. The width of the valley arc and the projection width of the burr are measured, as shown in Figure 7b. Calculating its arithmetic means, the results are presented in Table 4. For the texture unit, *k_s_* = 93.5% is derived from the definition of the area ratio of the retro-reflection. By selecting a high-precision machining system and analyzing and optimizing the cutting tools and cutting boundaries, the original mold obtained meets the retro-reflective requirements of the optical functional texture of the triangular pyramid.

## 4. Conclusions

(1) According to the precision and preparation process of the triangular pyramid optical functional texture film, technical requirements, including the shape, size and surface roughness of the original mold are determined. An ultra-precision machining system is used to both design the tool geometry according to the machining plan and optimize the cutting boundaries. Through the cutting experiment, an original mold with a surface roughness of Ra less than 10 nm and a surface morphology that satisfies the requirements of the optical functional texture is obtained.

(2) Low-speed forming planning technology is adopted in the ultra-precision machining system to avoid large fluctuations in the cutting force caused by the change in the cutting thickness, and the processing precision of the fine structure can be strengthened.

(3) As the independent arrangement of the unit structure on the surface functional texture of the part, the unit structure has intermittent cutting. There will be fine burrs at the cutting end of each unit structure. The effect of the burr on the functional texture surface quality can be effectively reduced by optimizing the cutting tool, cutting material and cutting parameters.

(4) The optical properties of the optical functional texture can be indirectly evaluated by analyzing the distribution of the edge burr and calculating the area ratio of the retro-reflection of the triangular pyramidal texture unit of the processed mold.

## Figures and Tables

**Figure 1 micromachines-11-00248-f001:**
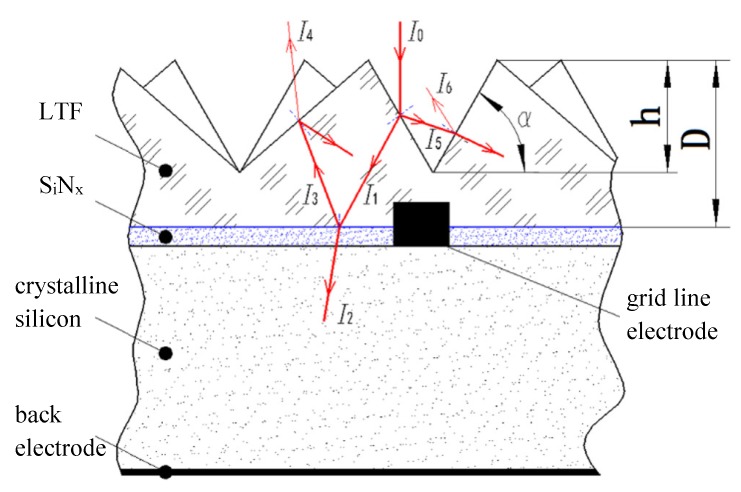
Composite laminate structure.

**Figure 2 micromachines-11-00248-f002:**
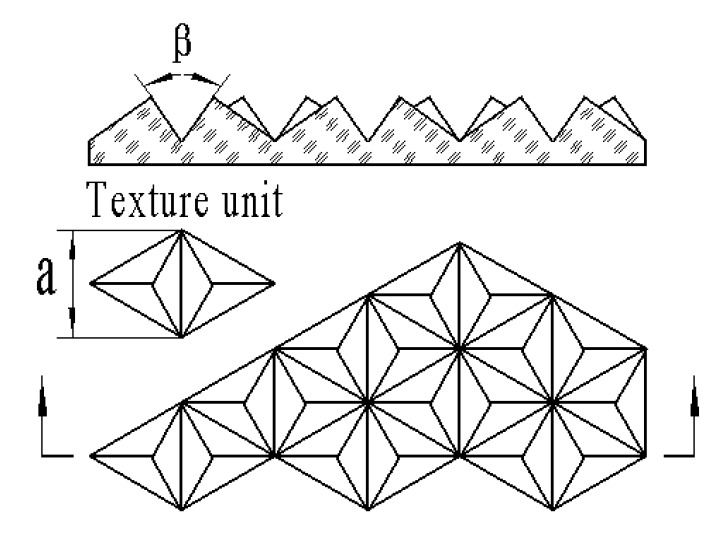
Triangular pyramid microstructure model.

**Figure 3 micromachines-11-00248-f003:**
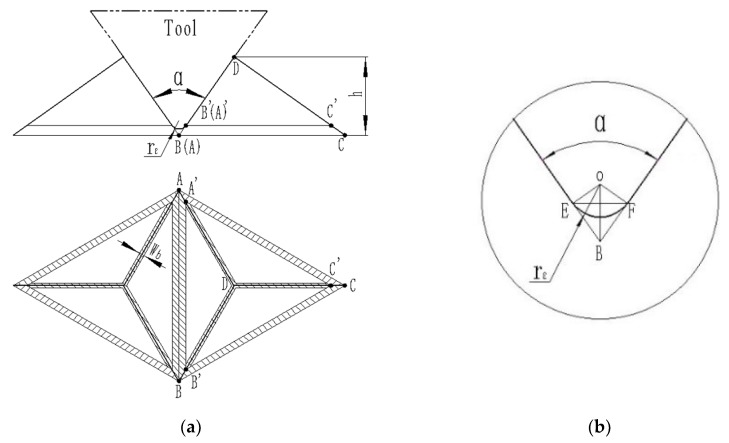
Invalid retro-reflection region: (**a**) Schematic diagram of the invalid retro-reflection area, (**b**) invalid area of the valley

**Figure 4 micromachines-11-00248-f004:**
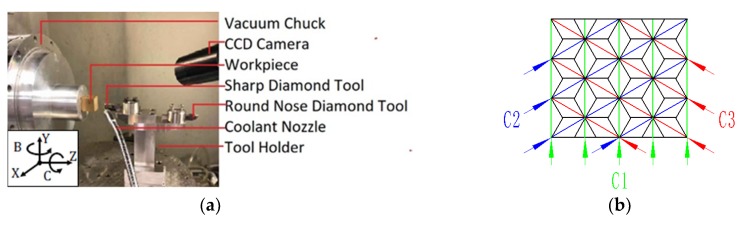
Ultra-precision single-crystal diamond cutting system: (**a**) Clamping mode of the workpiece and tool, (**b**) feed direction.

**Figure 5 micromachines-11-00248-f005:**
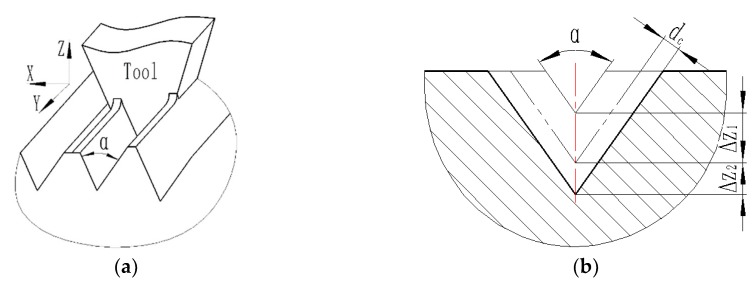
Stepped cutting in the Z-axis: (**a**) V-groove cutting schematic diagram, (**b**) Stepper deep feed.

**Figure 6 micromachines-11-00248-f006:**
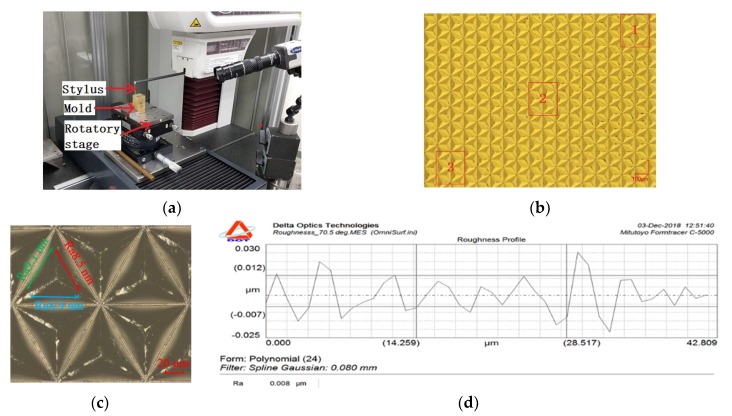
Surface roughness measured by a stylus profilometer: (**a**) Measuring conditions, (**b**) area of measurement, (**c**) measuring position, (**d**) roughness value.

**Figure 7 micromachines-11-00248-f007:**
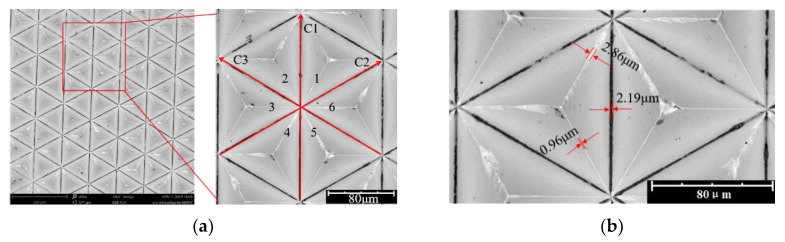
SEM (scanning electron microscope) of the burr and invalid zone: (**a**) Edge burrs, (**b**) Measurement of the invalid zone of the texture unit.

**Table 1 micromachines-11-00248-t001:** Tool geometry parameters.

Rake Angle (°)	Clearance Angle (°)	Included Angle (°)	Tool Nose Aadius (µm)	Cutting-Edge Radius (µm)
0	15	70.5	0.1	0.1

**Table 2 micromachines-11-00248-t002:** Material characteristics of the tool and workpiece.

Material	Young’s Modulus (GPa)	Poisson’s Ratio	Density (g/cm^3^)	Thermal Conductivity (W/m·K)	Specific Heat (J/(kg·K))	Hardness (HB)
Single diamond	960	0.2	3.5	2000	507.9	8000
Brass	105	0.324	8.4	104.7	374	69.3

**Table 3 micromachines-11-00248-t003:** Measurement of the triangular pyramid microstructure size and roughness.

Sampling Area	Triangular Pyramid Edge Length		Triangular Pyramid Height		Roughness (nm)
	Mean Value (µm)	Deviation	Mean Value (µm)	Deviation	
1	113.41	−1.38%	51.41	2.82%	9.5
2	114.51	−0.42%	49.62	−0.76%	8.9
3	113.41	−1.38%	50.74	1.48%	9.3
Mean value	113.78	1.06%	50.59	1.18%	9.2

**Table 4 micromachines-11-00248-t004:** Arc width of the valley bottom and burr projection width.

Sampling Area	Arc Width of the Valley Bottom EF	Edge Burr Width at the Cut-in	Edge Burr Width at the Cut-out
	Mean Value (µm)	Mean Value (µm)	Mean Value (µm)
1	2.75	0.96	2.13
2	2.64	0.97	2.08
3	2.74	0.97	2.11
Mean value	2.71	0.97	2.11

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
