# Peer review of "Precision Cutting of the Molds of an Optical Functional Texture Film with a Triangular Pyramid Texture"

_micromachines, 2020, doi:10.3390/mi11030248_

Round 1
Reviewer 1 Report
The authors present an interesting study about cutting of pyramid structures which can be possibly used in optical manipulation. Nevertheless, I have a few questions that need to be addressed:
(1) The dimension of pyramid fabricated by the current methods is limited to a few microns. Honestly, this might be too large for "functional optical structures" which are usually in the sub-wavelength range. Such pyramids structures in nanoscale can be possibly fabricated by chemical etching and lithography. I think the authors need to justify the novelty and motivation of the proposed method.
(2) I think it is necessary to provide real optical measurements to the function of fabricated structures. Although the fabrication process is interesting, it is more important to justify the motivation and reasons to do such study. The author needs to explain the targeted spectral range and potential applications.
Reviewer 2 Report
The paper titled "Precision cutting of the molds of an optical functional
3 texture film with a triangular pyramid texture" presents a method to cut optical films with machining tools. The content and subject of the paper may better be accommodated in other journals. Aims and scope of Micromachines is more towards micro/nano scale device applications, or novel micro/nano fabrication techniques. In its present form, this paper is more suitable for a journal that deals with machining of parts and machining tools.
Reviewer 3 Report
This paper reports the machining of a metal (brass) mold which contains triangular pyramid textures for the purpose of hot embossing of optical films. It is a good technical report but may not be a good scientific paper for lacking significant originality and novelty. There are some comments as follows.
Why the ks defined in Eq.(1) can be used to evaluate the optical performance of embossed optical films? The shapes of microstructures in the mold may deviate those in the mold. Any useful references? What kind of role does the Fy as defined in Eq.(2) play in this paper? How does it affect the machining processes? Who can you assure the third cut (C3) can be precisely coincident with the previous two cuts (C1 and C2) to form accurate triangular patterns? How much of the contribution obtained in this work is coming from the ultra-precision machining system itself as compared with your effort? Or, how can other people be benefited from this paper if their ultra-precision machining systems are different from yours?Author Response
Please see the attachment.

Round 2
Reviewer 1 Report
The manuscript has been revised and improved. I recommend acceptance of the current manuscript.
Reviewer 3 Report
The authors have pretty much answered all my questions in a satisfactory manner.
This manuscript is a resubmission of an earlier submission. The following is a list of the peer review reports and author responses from that submission.